# Clinical Validation of a Size-Based Microfluidic Device for Circulating Tumor Cell Isolation and Analysis in Renal Cell Carcinoma

**DOI:** 10.3390/ijms24098404

**Published:** 2023-05-07

**Authors:** Tito Palmela Leitão, Patrícia Corredeira, Sandra Kucharczak, Margarida Rodrigues, Paulina Piairo, Carolina Rodrigues, Patrícia Alves, Ana Martins Cavaco, Miguel Miranda, Marília Antunes, João Ferreira, José Palma Reis, Tomé Lopes, Lorena Diéguez, Luís Costa

**Affiliations:** 1Instituto de Medicina Molecular João Lobo Antunes, Faculdade de Medicina, Universidade de Lisboa, Av. Prof. Egas Moniz, 1649-028 Lisboa, Portugal; 2Faculdade de Medicina, Universidade de Lisboa, Av. Prof. Egas Moniz, 1649-028 Lisboa, Portugal; 3Urology Department, Hospital de Santa Maria, Centro Hospitalar Universitário Lisboa Norte, Av. Prof. Egas Moniz, 1649-028 Lisboa, Portugal; 4Department of Clinical and Molecular Medicine, Faculty of Medicine and Health Sciences, Norwegian University of Science and Technology, Erling Skjalgsons gate 1, 7491 Trondheim, Norway; 5Biological Engineering Department, Instituto Superior Técnico, Av. Rovisco Pais 1, 1049-001 Lisboa, Portugal; 6International Iberian Nanotechnology Laboratory, Avenida Mestre José Veiga s/n, 4715-330 Braga, Portugal; 7RUBYnanomed Lda, Praça Conde de Agrolongo 123, 4700-312 Braga, Portugal; 8CEAUL—Centro de Estatística e Aplicações, Faculdade de Ciências, Universidade de Lisboa, 1749-016 Lisboa, Portugal; 9Oncology Division, Hospital de Santa Maria, Centro Hospitalar Universitário Lisboa Norte, Av. Prof. Egas Moniz, 1649-028 Lisboa, Portugal

**Keywords:** circulating tumor cell, kidney cancer, liquid biopsy, microfluidic, renal cell carcinoma

## Abstract

Renal cell carcinoma (RCC) presents as metastatic disease in one third of cases. Research on circulating tumor cells (CTCs) and liquid biopsies is improving the understanding of RCC biology and metastases formation. However, a standardized, sensitive, specific, and cost-effective CTC detection technique is lacking. The use of platforms solely relying on epithelial markers is inappropriate in RCC due to the frequent epithelial-mesenchymal transition that CTCs undergo. This study aimed to test and clinically validate RUBYchip™, a microfluidic label-free CTC detection platform, in RCC patients. The average CTC capture efficiency of the device was 74.9% in spiking experiments using three different RCC cell lines. Clinical validation was performed in a cohort of 18 patients, eight non-metastatic (M0), five metastatic treatment-naïve (M1TN), and five metastatic progressing-under-treatment (M1TP). An average CTC detection rate of 77.8% was found and the average (range) total CTC count was 6.4 (0–27), 101.8 (0–255), and 3.2 (0–10), and the average mesenchymal CTC count (both single and clustered cells) was zero, 97.6 (0–255), and 0.2 (0–1) for M0, M1TN, and M1TP, respectively. CTC clusters were detected in 25% and 60% of M0 and M1TN patients, respectively. These results show that RUBYchip™ is an effective CTC detection platform in RCC.

## 1. Introduction

Kidney cancer (KC) is the 14th most common malignancy worldwide, with a global incidence of 431,288 in 2020 [1]. The incidence in Europe and North America is considerably higher than in other regions, ranging from 2.09 cases per 100,000 inhabitants (age-standardized rate) in Middle Africa to 24.7 in North America [2]. Renal cell carcinoma (RCC) accounts for the majority (90%) of KC cases [2]. The predominant histological subtypes are clear cell RCC (ccRCC; 70%), papillary RCC (pRCC; 10–15%), and chromophobe RCC (cRCC; 5%) [2]. The remaining histological subtypes account for less than 1% each [2]. About one-third of RCC patients are diagnosed in metastatic stage and up to 40% of those treated with curative intent relapse and develop metastases during follow-up [3,4].

A clinically useful biomarker is missing for RCC, as diagnosis and follow-up still rely solely on cross-sectional imaging. Several potential biomarkers have been investigated, but none has shown the accuracy and ease of use required for clinical application, particularly for guiding disease management.

The focus of the current research on RCC biomarkers is liquid biopsy. The principle underlying this method is obtaining tumor-derived biological material circulating in the bloodstream through a simple blood sample and accessing phenotypic and genetic data of primary and secondary tumors without the invasiveness of a tumor or metastasis biopsy. This can allow minimally invasive early cancer diagnosis and repeated sequential sampling during disease management to accurately guide treatment decisions, monitor treatment response, and provide prognostic information. Liquid biopsy can focus on a multitude of circulating biomarkers, including circulating tumor DNA (ctDNA), micro RNA (miRNA), and circulating tumor cells (CTCs) [5,6,7,8].

CTCs have been a particular focus of interest in liquid biopsy research. The potential clinical value of CTCs has been explored in several tumor types, such as breast, colon, and prostate [9]. Five studies have found a correlation between the presence of CTCs and CTC counts and prognostic outcome measures in RCC, despite their short follow-up and the fact that outcome measures were not primary endpoints [10,11,12,13,14]. Vimentin-expressing CTCs also seem to correlate with more advanced RCC stages [15].

The scientific community is still searching for a sensitive, specific, reproducible, and cost-effective CTC detection technique. CTCs are extremely rare compared to whole blood cells, with estimates indicating one CTC per billion normal blood cells in metastatic disease [16]. CTC enrichment techniques can be classified in four categories: antibody-based (immunomagnetic beads or microfluidics), density-based, size-based (microfluidics and membrane filters), and electrophoresis-based [4]. CTC detection and identification has been accomplished through five different techniques: immunocytochemistry, reverse transcriptase-polymerase chain reaction (RT-PCR), cytomorphological criteria, flow cytometry with immunofluorescence, and fluorescence in-situ hybridization (FISH) [4]. Cellsearch^®^ is currently the main CTC detection platform approved by the Food and Drug Administration (FDA) for clinical use and relies on immunomagnetic enrichment and fluorescent labeling for CTC detection [17]. However, it is deemed inappropriate for use in RCC due to the lack of epithelial cell adhesion molecule (EpCAM) and cytokeratin (CK) expression in CTCs that have undergone epithelial-mesenchymal transition (EMT), a very common phenomenon in this type of tumor [15]. Only 18.6% of RCC CTCs express EpCAM [15], which explains the particularly low CTC detection rates achieved in initial studies with this marker [18,19].

CTC isolation based on their physical properties is a simpler and more efficient method that relies on differences in cell size and deformability and is independent of molecular markers. However, although more sensitive, it may have lower specificity, given CTC heterogeneity [20]. The principle underlying size-based CTC isolation platforms is the larger size and lower deformability of CTCs compared to blood cells [4]. They have the advantage of minimizing cell loss and allowing downstream analysis of intact cells, but also the limitations of device clogging and some loss of CTCs that are smaller than the device pores. Several authors have used size-based CTC isolation in RCC patients, both membrane-based [21,22,23,24,25,26,27,28] and microfluidic devices [29,30,31]. Most of these devices are designed to capture cells larger than 8 μm, letting both smaller erythrocytes and deformable leukocytes pass through [32]. Microfluidic-based devices minimize sample processing steps and require shorter processing times, since no sample pre-processing is required [33]. They also require lower reagent volumes and have low contamination issues and sample loss rates. Therefore, cell loss is minimized, especially in samples with low CTC concentration, usually yielding higher sensitivity and detection rates [4].

Due to their potential advantages, microfluidic devices have been investigated as a promising CTC isolation method. RUBYchip™ was shown to be significantly superior to the FDA-approved CellSearch^Ⓡ^ in CTC isolation in breast, colorectal, gastric, and pancreatic cancer [34,35,36].

The aim of this study was to test and validate the RUBYchip™ microfluidic device for CTC isolation and analysis in RCC.

## 2. Results

### 2.1. CTC Isolation Efficiency

The spiking experiments performed with the RUBYchip™ device resulted in a high RCC CTC yield. The optimal overall capture efficiency was obtained at a 80 μL/min flow rate, enabling the isolation of 77.7%, 77.2%, and 69.8% of spiked Caki-2, A-498, and 786-O cells, respectively (Figure 1). The mean capture efficiency for all RCC cell lines analyzed was 74.9%. At 100 and 120 μL/min, capture efficiency was lower and with higher variability between cell lines, and hence patient samples were subsequently processed at 80 μL/min flow rate.

### 2.2. Characteristics of the Study Cohort

The clinicopathological characteristics of the study’s patient cohort are summarized in Table 1 (clinical patient database in Appendix A). The median age at diagnosis was 60 years for the M0 and M1TP groups and 71 years for the M1TN group. Overall, eight patients (72.7%) had ccRCC, one patient (9.1%) had cRCC, and two patients (18.2%) had pRCC.

### 2.3. CTC Counts and Characterization

CTCs were detected either as single cells or as clusters of two or more cells (Figure 2). The total CTC count refers to the sum of single CTCs and the number of CTCs in clusters. Two CTC phenotypes were found in all patient groups: epithelial and mesenchymal CTCs. Of note, no EMT CTCs were detected in this cohort.

Table 2 and Figure 3 depict CTC counts stratified by phenotype and patient group (M0, M1TN, and M1TP). The detection rate was 77.8% (14/18) overall, 75.0% (6/8) in M0 group, and 80.0% (8/10) in M1 group. CTC clusters were detected in 25.0% (2/8) of M0 patients and in 60.0% (3/5) of M1TN patients. Interestingly, no CTC clusters were detected in M1TP patients. All CTC clusters were composed only of mesenchymal cells.

The average (range) total CTC count was 6.4 (0–27), 101.8 (0–255), and 3.2 (0–10) in M0, M1TN, and M1TP groups, respectively. M1TN patients showed a significantly higher number of CTCs than M1TP counterparts (31.8 times higher on average; *p* = 0.0003, 90% CI 30.0–345.6; Figure 4), a difference mainly attributed to the presence of mesenchymal CTCs. The average (range) total (single + clustered) mesenchymal CTC count was 3.1 (0–21), 97.6 (0–255), and 0.2 (0–1) in M0, M1TN, and M1TP patients, respectively, with M1TN patients having significantly more total mesenchymal CTCs than M1TP patients (488.0 times more on average; *p* < 0.0001, 90% CI 31.7–7510.5).

M1TN patients showed significantly more single mesenchymal CTCs than either M0 (*p* = 0.007) or M1TP (*p* < 0.001) patients. The M1TN group had an average (range) of 59.6 (0–157) single mesenchymal CTCs compared to 0.2 (0–1) in the M1TP group, which means that M1TN patients had 297 times more single mesenchymal CTCs on average than M1TP patients (*p* < 0.001, 90% CI 20.9–4231). Interestingly, no single mesenchymal CTCs were found in the M0 group, although mesenchymal CTCs in clusters were present.

The average (range) single CTC count was 3.3 (0–13), 63.8 (0–157), and 3.2 (0–10) in M0, M1TN, and M1TP groups, respectively. M1TN patients had significantly more single CTCs than M1TP patients (63.8 times more on average; *p* = 0.0012, 90% CI 19.8–205.3). Although more clusters were detected in the M1TN compared to the M0 group, this difference was not statistically significant.

Patients under antiplatelet therapy had significantly more single CTCs (*p* = 0.025), total CTCs (*p* = 0.029), and mesenchymal clusters (*p* = 0.031) compared to patients not receiving that therapy (Figure 5).

### 2.4. Correlation of Clinical Variables with CTC Count and Phenotype

Despite the small number of samples assessed, a strong positive correlation was found between CTC counts and international normalized ratio (INR) in both M0 and M1 groups. In the M0 group, INR correlated with mesenchymal CTCs in clusters and total mesenchymal CTCs (r = 0.85, *p* = 0.008 and r = 0.85, *p* = 0.008, respectively). In the M1 group, INR correlated with mesenchymal CTCs (in clusters r = 0.970, *p* = 0.001, single r = 0.969, *p* = 0.002, and total r = 0.970, *p* = 0.002), with CTC clusters (r = 0.975, *p* = 0.001), with single CTCs (r = 0.974, *p* = 0.001), and with total CTCs (r = 0.973, *p* = 0.001).

Interestingly, in the M0 group, a negative correlation was found between epithelial CTCs and leucocyte count (r = −0.748, *p* = 0.033), while in the M1 group that correlation was positive (r = 0.80, *p* = 0.005).

In the M0 group, a strong correlation was found between weight and mesenchymal CTCs (r = 0.828, *p* = 0.011 and r = 0.828, *p* = 0.011 for mesenchymal CTCs in clusters and total mesenchymal CTCs, respectively) and between body mass index (BMI) and mesenchymal CTCs (r = 0.878, *p* = 0.004 for both mesenchymal CTCs in clusters and total mesenchymal CTCs).

In the M1 group, increased leukocyte and neutrophil counts also strongly correlated with epithelial CTCs (r = 0.801, *p* = 0.005 and r = 0.852, *p* = 0.002, respectively). A moderate positive correlation was found in the entire cohort and in M1 group between neutrophil-to-lymphocyte ratio (NLR) and total CTC counts, single CTCs, mesenchymal CTCs, and CTCs in clusters.

In M1 patients, serum platelet counts moderately and inversely correlated with total CTCs (r = −0.714, *p* = 0.0203), single CTCs (r = −0.713, *p* = 0.0206), CTC clusters (r = −0.715, *p* = 0.0201), and mesenchymal CTCs in clusters (r = −0.714, *p* = 0.0203), but the same was not observed in M0 patients.

Serum albumin level was moderately and inversely correlated with epithelial CTC counts in M1 patients (r = −0.83, *p* = 0.01), while serum hemoglobin level was moderately correlated with total CTC counts in M0 (r = −0.747, *p* = 0.033) but not in M1 group.

Figure 6 depicts the correlations found in this study between CTC counts and clinical variables assessed.

The correlation plots can be found on Appendix A. 

### 2.5. Survival Analysis

Survival analysis is shown in Figure 7. Patients with 5 or more total CTCs had a decreased OS compared to patients with less than 5 CTCs (hazard ratio [HR] 8.45, 95% CI 1.29–55.22; *p* = 0.0143; Figure 7A). This was also true for patients with 5 or more single and total mesenchymal CTCs (HR 7.657, 95% CI 0.717–81.78; *p* = 0.0044; Figure 7C,D) and for those with CTC clusters (HR 0.1306, 95% CI 0.012–1.395; *p* = 0.008). INR, BMI, and NLR did not impact OS. The median follow-up was 11.2 months.

## 3. Discussion

### 3.1. Detection Rates

The mean capture efficiency of RCC cells in RUBYchip™ at the optimal flow rate of 80 μL/min was 74.9%. This high efficiency may be attributed to the absence of whole blood sample preprocessing and to the chip design with prefilters to prevent microclots and obstruction and microfilters to enable white blood cell clearance [36]. The microfilter geometry allows a good balance between the free passage of smaller and/or more deformable cells, like blood cells, and the entrapment of larger and less deformable cells, like CTCs. CTCs are usually larger and less deformable, due to their large nuclei and high cytoplasm-to-nucleus ratio [38]. The wide variability of CTC sizes and phenotypes [39] has prompted the use of three RCC cell culture lines in this study. The results obtained showed consistent detection rates among the three cell lines at the determined optimal flow rate. 

In a preclinical validation study of RUBYchip™ conducted in metastatic breast cancer, CTC capture efficiency with this device was up to 10 times higher compared to the CellSearch^®^ system [36]. This was probably because CellSearch^®^ uses preservation tubes, pre-processes blood samples, and only targets EpCAM + CTCs, leading to CTC loss during sample processing. 

CTC detection rate in this study’s RCC patient cohort was 77.8% overall, reaching 80% in M1 patients and 75% in M0 patients. This is a high detection rate compared to the median of 57% (interquartile range [IQR] 55%) found in a previous systematic review of our group on CTC detection techniques in RCC [4]. RCC is known to have lower CTC detection rates compared to other tumor types, which is thought to be due to a greater prevalence of EMT in this tumor and consequent loss of epithelial markers standardly used to identify these cells [4]. CellSearch^®^ is the first technology approved by the FDA for CTC detection and is regarded as the benchmark in most epithelial cancers, except RCC [40]. It is documented that not all CTCs express EpCAM [41,42], and in the case of RCC, only 18.6% of CTCs seem to express this marker [15]. 

The similar CTC detection rate achieved in localized and metastatic disease is an interesting finding of this study, as it indicates that most cancer patients have CTCs, even in localized disease stages. Rather than the proportion of patients with CTCs, it is the CTC count that seems to vary with disease stage [4]. 

### 3.2. CTC Count

M1TN patients were found to have significantly higher total CTC, single CTC, mesenchymal CTC, and total mesenchymal CTC counts compared to M1TP patients. M1TN patients had 31.8 and 15.9 times more total CTCs than M1TP (*p* = 0.0003) and M0 patients, respectively. These differences are substantial and mainly attributed to the increase in mesenchymal CTCs, with M1TN patients showing 488 and 31 times more total mesenchymal CTCs than M1TP (*p* < 0.0001) and M0 patients, respectively.

Compared to M0 patients, M1TN patients also showed significantly higher total CTC, single CTC, mesenchymal CTC, and total mesenchymal CTC counts, although statistical significance was only achieved for single mesenchymal CTCs (*p* = 0.007). 

These findings are in line with other reports in the literature. One other study also reported higher CTC counts in patients with metastatic compared to localized RCC (9.6 vs. 5.3 CTC/7.5 mL) [43]. Liu S. et al. found that CTC counts were 2.2 times increased in late (3 and 4) compared to early (1 and 2) disease stages (*p* < 0.001) [15]. The same authors correlated mesenchymal CTCs with RCC stage. Several other studies have demonstrated a correlation between CTC presence and staging, particularly with N+ and M+ status [10,15,22,29,43,44,45,46].

Epithelial CTC counts were very similar among groups, suggesting that disease stage does not have an impact on their number. Interestingly, no EMT CTCs were found in the study cohort. This can be due to the early disease stage of M0 patients, whose cancer cells may not have yet undergone EMT, and/or to the very advanced disease stage of M1 patients, whose transitioning cancer cells may have undergone full epithelial marker loss and concomitant gain of mesenchymal markers, like vimentin.

No significant differences were found in CTC counts between M1TP and M0 patients. This can be a marker of efficacy of systemic therapies in disease control and in limiting CTC release, despite the observed clinical progression. Additionally, no differences were found in CTC counts according to patient characteristics like N stage, smoking, obesity, hypertension, or diabetes. On the other hand, this study was underpowered for the analysis of ECOG score, T stage, and tumor histological subtypes.

Most CTCs in the M1TN group were found to be mesenchymal and only one M1 patient presented epithelial CTCs, which is in line with the relevant EMT known to occur in advanced RCC. Although the metastatic process is not yet fully understood, it is generally accepted that EMT plays a role in CTC release and is an important factor explaining tumor progression and treatment resistance [47]. A link between EMT and disease aggressiveness, treatment response, and survival has already been established in several tumor types [48,49]. 

In this study, mesenchymal CTCs were defined as DAPI+/CD45-/CK-/Vim+ cells. However, some controversy exists in this definition. Vimentin is the most used marker for mesenchymal phenotyping, but other markers, like N-cadherin, O-cadherin, fibronectin, serpin peptidase inhibitor, and twist have been studied [50]. However, no marker or panel of markers has been identified as being able to definitely identify EMT or mesenchymal CTCs to date. Further characterization of these CTC subsets, for instance with downstream analysis using DNA and RNA sequencing, may improve the understanding of EMT and the role of these cells in cancer progression [51]. It has been proposed that some vimentin-positive cells could be circulating cancer-associated fibroblasts (cCAF) [52]. It has also been reported that metastatic CTCs are more viable when integrated into heterotypic clusters consisting of tumor and stromal cells [53]. Spindle-shaped vimentin-positive cells were identified in some samples in this study and considered as possible cCAF and not counted as CTCs. Only vimentin-positive CTCs with specific cytomorphological features, like round/ovoid shape, big nuclei, and high nucleus-to-cytoplasm ratio have been considered.

It should be noted that the first FDA-approved technology for CTC capture and analysis in clinical setting in most tumors—CellSearch^®^—only relies on the expression of epithelial markers, which has hindered RCC CTC research in the context of the widely present EMT in this tumor type. Hence, CTC isolation platforms capable of detecting EMT, mesenchymal CTCs, and CTC clusters, like the one used in this study, should be employed in future RCC research to clarify the clinical significance of these CTC subpopulations, and further elucidate the biology of kidney cancer. In 2022, the Parsortix^®^ microfluidic platform has also received FDA approval for CTC detection in metastatic breast cancer, confirming microfluidics has a promising technology in cancer. 

### 3.3. CTC Clusters 

CTC clusters can be defined as a group of 2–3 to 100 cancer cells [50,54]. Aceto et al. reported that CTC clusters had 23–50 times more metastatic potential than single CTCs, despite representing only 2–5% of all CTC events detected in a breast cancer mouse model [54]. Animal models have demonstrated that CTC clusters arise from primary tumor vein invasion and fragmentation rather than aggregation of single CTCs [55]. It has also been shown that the injection of clustered cells resulted in reduced OS in mice compared to the injection of single CTCs (12.7 vs. 15.7 weeks, *p* < 0.016) [54]. In the same mouse model, CTC clusters were cleared from circulation at least three times more rapidly than single CTCs (half-life: 6–10 min for CTC clusters vs. 25–30 min for single CTCs) [54].

In the present RCC patient cohort, a higher average CTC cluster count was found in M1TN compared to M0 and M1TP patients. Although these differences were not statistically significant (possibly due to the small sample size), they seem clinically relevant, particularly in the M1TP group, where no clusters were found, suggesting efficacy of systemic treatment in preventing CTC cluster formation and release. Interestingly, all CTCs in these clusters were mesenchymal, in agreement with other data in the literature reporting that CTC clusters seem to be more frequently composed of mesenchymal rather than epithelial CTCs [51].

### 3.4. CTCs and Survival Outcomes

Patients with 5 or more total CTCs, with mesenchymal CTCs (both single and total), and with CTC clusters were found to have significantly lower OS in this study. Several other studies had previously documented the impact of the presence of CTCs and of CTC counts on RCC survival [4,10,12,13,56,57,58]. One study showed that patients with CTC counts with >0.12 CTCs/mL annually, had shorter OS compared to patients with <0.12 CTCs/mL (median 17.0 vs. 21.1 months, *p* < 0.001) [56]. In another study, patients with mesenchymal CTCs had a slight survival decrease compared to patients without these cells (HR 1.2, 95% CI 1.1–1.4; *p* = 0.005) [59]. Another group found that total postoperative CTC counts higher than 6, presence of postoperative mesenchymal CTCs, and presence of postoperative CTC-white blood cell clusters significantly correlated with recurrence and metastases [57]. In a study in M1 RCC, patients with total CTC counts >3 had shorter OS than patients with ≤3 (median 13.8 vs. 52.8 months on multivariate analysis; HR 1.67, 95% CI 0.95–2.93; *p* = 0.003) [13]. A standard CTC count cut-off to predict prognostic outcomes is yet to be determined.

Similar findings were reported in other tumor types. In colorectal cancer, patients with more than 3 CTCs/7.5 mL had reduced survival [60,61]. The risk of tumor progression and death was higher in CTC-positive patients with pancreatic and esophageal cancer [62,63]. OS also correlated with high CTC counts in a meta-analysis of gastric cancer [64].

### 3.5. Correlation of CTC Counts with Clinical Variables

A very strong positive correlation was found between all CTC counts and INR in metastatic patients, with the correlation being more moderate in patients with localized disease. This could be explained by the acknowledged prothrombotic state elicited by CTCs and cancer in general, which may cause coagulation factor consumption and increased INR [65]. The EMT process can cause overexpression of tissue-factor in CTCs, conferring procoagulant properties that can contribute to metastases formation [65,66]. However, in the study by Dirix and colleagues, no significant association was found between activated partial thromboplastin clotting time or prothrombin time and CTC counts [67]. 

Patients’ weight and BMI showed a strong positive correlation with mesenchymal CTC counts in the M0 group. Age showed a moderate positive correlation, consistent with all CTC counts. It can be hypothesized that age may hinder immunity, which, together with the general patient frailty, may promote CTC survival.

Since CTCs in circulation interact with other blood cells and components, a possible relation between CTC counts and other blood constituents was investigated. 

Leukocyte counts were found to have a moderate positive correlation with epithelial CTCs in M1 patients, but a negative correlation in M0 patients. The latter was only observed for epithelial CTCs and not for mesenchymal CTCs or clusters. Tumor-associated neutrophils appear to contribute to CTC survival by suppressing peripheral leukocyte activation in advanced cancer patients [68]. Additionally, single CTCs have been shown to have impaired interactions with T lymphocytes and natural killer (NK) cells, being this way protected against recognition by the immune system [69]. On the other hand, heterotypic CTC clusters have increased aggressiveness, namely when CTCs are conjugated with platelets, leukocytes, neutrophils, tumor-associated macrophages, and fibroblasts [69]. In addition, metastasis-promoting gene expression profile changes were shown to occur with CTC and neutrophil interaction in a breast cancer mouse model study [69]. 

NLR positively correlated with total CTCs, single CTCs, mesenchymal CTCs, and CTC clusters in this cohort, particularly in metastatic patients. In M1 group, increased neutrophil counts also strongly correlated with epithelial CTCs, but not with other CTC variables. In a study by Peyton et al., an increase in absolute neutrophil count and NLR >4 were independent predictors of decreased survival in RCC (*p* < 0.05) [70]. In stage II/IV gastric cancer, CTC detection was also significantly correlated with neutrophil count (*p* = 0.020) and NLR (*p* = 0.009) [71]. Therefore, NLR and neutrophil counts may prove to be surrogate predictors of survival in RCC.

In the present study, C-reactive protein (CRP) levels positively correlated with epithelial CTC counts, suggesting that an elevation in inflammatory parameters may correlate with increased CTC count. In a study in ovarian cancer, CRP was higher in CTC-positive versus -negative patients, with a median of 4.33 (IQR 1.46–7.51) versus 1.52 (IQR 0.50–4.50), respectively (*p* = 0.001) [72]. CRP has been shown to have prognostic value in predicting outcomes, as well as the ability to predict response to chemotherapy in various tumor types [73]. A recent study demonstrated a strong correlation in RCC between coagulation and both CRP and CTCs [74]. 

In this study’s cohort, serum platelet count was inversely correlated with total CTC, single CTC, and CTC cluster counts, but only in metastatic patients. Some studies have suggested that activated platelets can shield CTCs and protect them from immune destruction and blood flow shear forces [75,76,77]. CTC-coating platelets can produce major histocompatibility complex I-positive vesicles that may help CTCs to escape recognition by NK and T cells [78]. This platelet recruitment and activation could lead to platelet consumption, decreasing their serum counts, which would help explain the inverse relation found in this study. A 2022 paper by Dirix et al. also found a negative correlation between platelet count and CTC count in advanced breast cancer (*p* < 0.0009, R2 0.167) [67]. On the other hand, Guan et al. found a positive correlation between mesenchymal CTCs and platelet levels in RCC [59].

Platelet interaction with CTCs may also lead to EMT induction and maintenance through TGF-ß release, thereby promoting metastases formation [79]. This suggests that platelet action may promote CTC survival and metastases, but further studies are required to clarify the relation between platelet and CTC counts.

Serum hemoglobin levels were found to correlate with total CTC counts moderately and inversely in the M0 group. In a study in prostate cancer, a negative association was found between CTC counts and hemoglobin levels (*p* = 0.004) [80], and other studies have confirmed this association [81,82]. Also, serum albumin showed a moderate and inverse correlation with CTC counts in M0 patients, but only with epithelial CTC counts. In this cohort, patients with more advanced disease and higher tumor burden had poorer performance status and concomitantly lower levels of albumin and higher levels of CRP. Hypoalbuminemia is a surrogate marker of known disease processes present in advanced cancers, like increased catabolism, systemic inflammatory response, increased vascular permeability and interstitial edema, and decreased albumin synthesis [83]. The correlation of serum albumin and CTCs is poorly studied. In a study in ovarian cancer, no differences were found in serum albumin levels between CTC+ and CTC- patients [72]. 

### 3.6. Final Remarks, Study Limitations and Future Directions

In sum, this study showed that the RUBYchip^TM^ consistently detected CTCs in distinct groups of patients suffering from RCC irrespective of CTC counts (low vs. high), phenotype (mesenchymal vs. epithelial) and degree of aggregation (singlets vs. clusters). In treatment-naïve patients with metastatic disease (M1TN group) we consistently found increased total CTCs, namely single CTCs, chiefly contributed by the mesenchymal phenotype.

The main limitation of this study is its small sample size, which made it underpowered for several analyses. Despite the limited clinical conclusions that can be drawn from such a small patient cohort, a positive correlation was identified between CTC counts and both staging and prognosis. 

The future of liquid biopsy in cancer is promising, being generally agreed that they will play a crucial role in cancer diagnosis, treatment, and monitoring in upcoming years. Some of the advantages of liquid biopsies are its non-invasive nature, real-time monitoring potential, and ability to provide a comprehensive picture of cancer cells and their behavior.

However, there are still challenges ahead, such as the need for standardization and improved accuracy of liquid biopsy testing. Nevertheless, research in the field is ongoing, and it is predictable that liquid biopsy technology will continue to move forward and become an increasingly important tool in the fight against cancer.

Further investigation is required to identify effective molecular markers and develop reliable, standardized techniques for isolation and detection of CTCs in RCC, so that they can be used as diagnostic, prognostic, and treatment management tools.

## 4. Materials and Methods

### 4.1. Microfluidic Device

The RUBYchip™ device (RUBYnanomed/International Iberian Nanotechnology Laboratory [INL], Braga, Portugal, PCT/EP2016/078406) is a microfluidic system that captures CTCs from whole blood samples based on cell size and deformability [36]. The device consists of an inlet that directs the sample through a network of interconnected capillaries into multiple cell-filtering chambers. Each chamber has transverse rows of micropillars that make up the cell filtering area. The size, geometry, and gap size of the pillars were designed so that deformable white blood cells gently flow through, while larger, more rigid cells, like CTCs, are retained in the cell-filtering chamber. The fabrication process, technical specifications, and details of the device are described elsewhere [36].

### 4.2. Cell Culture

Human KC cell lines Caki-2 (ATCC, HTB-47), A-498, and 786-O were used for spiking experiments. The Caki-2 cell line was cultured in McCoy’s 5A Medium (Gibco^TM^, Thermo Fisher Scientific Inc., Waltham, Massachusetts, USA), and the A-498 cell line in Dulbecco’s Modified Eagle Medium (Gibco^TM^), and 786-O in Roswell Park Memorial Institute 1640 (Gibco^TM^). All growth media were supplemented with 10% fetal bovine serum (Gibco^TM^) and 1% penicillin/streptomycin (Gibco^TM^). All cell lines were maintained at 37 °C with 5% CO_2_ in a humidified atmosphere, at a low passage, and routinely tested for mycoplasma contamination by quantitative polymerase chain reaction (GATC Biotech, Konstanz, Germany).

### 4.3. Spiking Experiments

The capture efficiency of the RUBYchip^TM^ device in RCC was assessed through spiking experiments using Caki-2, A-498 and 786-O cell lines. Approximately 200 cells were labeled with 4′,6-diamidino-2-phenylindole (DAPI) (10 μg/mL, Sigma-Aldrich, Burlington, MA, USA) after trypsinization and added to 7.5 mL of whole blood samples from healthy donors. To find the best conditions, samples were injected in RUBYchip™ using a syringe pump (KF Technology) at three different flow rates: 80, 100, and 120 μL/min. Afterwards, devices were washed with 2% bovine serum albumin (BSA) (NZYtech Lda, Lisbon, Portugal) in phosphate-buffered saline (PBS) 1X, fixed with 4% formalin for 20 min at room temperature and finally washed with 0.5% BSA in PBS 1X followed by 1% sodium azide (Sigma Aldrich) in PBS 1X. As previously described, cell counting control of the spiked cell number was performed by pipetting the same cell suspension volume into a well plate [36]. Fluorescence cell images were acquired using an inverted fluorescence Nikon Eclipse Ti microscope at 20× magnification. Experiments were performed in triplicate. Capture efficiency was calculated as the ratio between the number of DAPI-positive cells trapped inside the device and the average cell count in the well plate, as previously described by Ribeiro-Samy S, et al. [35]:(1)$$CTC\;capture\;efficiency\;\left(\%\right)=\frac{Trapped\;cells}{Spiked\;cells}\times 100\;\left(\%\right)$$

### 4.4. Immunocytochemistry Protocol and Immunofluorescence Imaging

Different experimental conditions were tested to optimize the antibody staining protocol in the cells trapped in the spiking experiments. The selected antibody panel included AF647 anti-human vimentin (Biolegend, 1:50, San Diego, CA, USA), phycoerythrin-conjugated anti-human CD45 (Invitrogen, Thermo Fisher Scientific, Waltham, MA, USA, 1:50), and DAPI (1 μg/mL). Two antibodies were tested to stain cytokeratin: anti-human cytokeratin 8/18 unconjugated ready-to-use antibody (Dako, Agilent, 200 μL, Santa Clara, CA, USA), detected with FITC goat anti-rabbit IgG cross-adsorbed secondary antibody (Invitrogen, Thermo Fisher Scientific, 1:1000); and FITC-conjugated anti-human cytokeratin (Miltenyi Biotec, Bergisch Gladbach, North Rhine-Westphalia, Germany). After isolation, cells were permeabilized with 0.25% triton X-100 for 10 min, then rinsed with PBS 1X. The antibody panel was incubated for 1 h at room temperature in the dark after a blocking step with 2% BSA in PBS 1X for 20 min. Samples incubated with unconjugated cytokeratin antibody were subsequently incubated with the secondary antibody for 30 min at room temperature in the dark and washed with 0.5% BSA in PBS 1X and 1% sodium azide in PBS 1X. Images were obtained using an inverted fluorescence Nikon Eclipse Ti microscope at 20× magnification.

### 4.5. Patient Recruitment and Sample Collection 

To validate RUBYchip™ for clinical use in RCC, 18 patients were enrolled at Centro Hospitalar Universitário Lisboa Norte (CHULN), Lisbon, Portugal, between August 2021 and May 2022. Patients were divided into three groups: a localized disease group (M0 group) with eight patients, whose samples were collected prior to treatment with curative intent; and a metastatic disease group (M1 group) of 10 patients, five of which were treatment naïve (M1TN group; n = 5,) and the remaining five were diagnosed with disease progression under systemic therapy (M1TP group; n = 5). Progression was defined according to RECIST criteria [84]. The study was approved by the Ethics Committee of CHULN and conducted in accordance with the Declaration of Helsinki and good clinical practice guidelines. All patients provided and signed informed consent before any study procedure. Objective tumor status was assessed through Tumor, Node, Metastasis (TNM) criteria [37]. Single 7.5-mL peripheral blood samples were collected in EDTA tubes at the time of diagnosis in M0 group, before systemic therapy in M1TN group, and after tumor progression in cross-sectional imaging follow-up in M1TP group. All samples were anonymized, and a code was assigned before sample processing.

### 4.6. CTC Isolation and Characterization

Clinical samples were processed in the RUBYchipTM device within 40 min after collection. The 7.5-mL blood samples were injected at 80 μL/min, and the CTCs entrapped in the device were then washed and fixed, as described in the spiking experiments (Section 2.3). CTCs were stained inside the device with the previously described antibody panel (Section 2.4) and under the same conditions. Samples from the first 12 patients were stained with the anti-human unconjugated cytokeratin 8/18, and those from the last six patients with FITC-conjugated anti-human cytokeratin 8/18.

After image acquisition, cells were manually enumerated and classified by randomized blind analysis performed by two independent operators. No variability was found in the cytokeratin signal with the two antibodies, but a reduction in the background FITC signal was observed with the conjugated antibody. Cells were identified as CTCs and distinguished according to their phenotype using the following criteria: epithelial CTCs were DAPI+/CD45-/CK+/Vim-, mesenchymal CTCs were DAPI+/CD45-/CK-/Vim+, and EMT CTCs were DAPI+/CD45-/CK+/Vim+ [35,36,85]. In addition, cells had to show membrane integrity in brightfield, a round nucleus, and cell-like morphology to be classified as CTCs. CTC clusters were defined as groups of two or more cells characterized by having regular contours, being in close contact with each other, and complying with the above criteria [54]. A CTC counting matrix was used to enumerate the difference CTC subsets (Appendix A)

### 4.7. Statistical Analysis

Continuous variables were presented as median, average, and range, and categorical variables as absolute and relative frequencies. Fisher’s exact test was used to compare the three patient groups (M0, M1TN, and M1TP) for the presence/absence of CTCs. Negative binomial regression was used to compare groups regarding CTC counts, including M1TN versus M1TP groups, pathological lymph node presence, smoking habits, hypertension, diabetes, overweight/obesity, and antiplatelet therapy. Pearson’s correlation was used to assess the correlation between CTC counts and quantitative clinical variables. Overall survival (OS) was defined as the time from sample collection to metastatic patients’ (M1TN and M1TP) last clinical follow-up or death and estimated using the Kaplan-Meier estimator. Median OS and 95% confidence interval (CI) were computed. Results were considered statistically significant if the *p*-value was less than 0.05. Statistical analyses were performed with R Software v2022.07.1 (R Foundation for Statistical Computing, Vienna, Austria). Survival analysis was performed using GraphPad Prism 8 v8.4.3 (686).

## 5. Conclusions

The findings of this study show that the RUBYchip™ microfluidic size-based CTC detection device is an effective and reproducible method for isolating CTCs in RCC. It has high detection rates with short processing times due to fewer processing steps compared to other devices. It is able to identify different CTC phenotypes and detect CTC clusters, which are relevant in this tumor type. The RUBYchip™ can thus be used in future RCC research to help improve the understanding of the metastatic process and disease progression, as well as to potentially guide patient management.

## 6. Patents

The RUBYchip™ is based on patent PCT/EP2016/078406, filed by INL in front of EPO on 22 November 2016, covering the geometry and surface coating of the microfluidic system for CTC isolation, and currently licensed exclusively to RUBYnanomed.

## Figures and Tables

**Figure 1 ijms-24-08404-f001:**
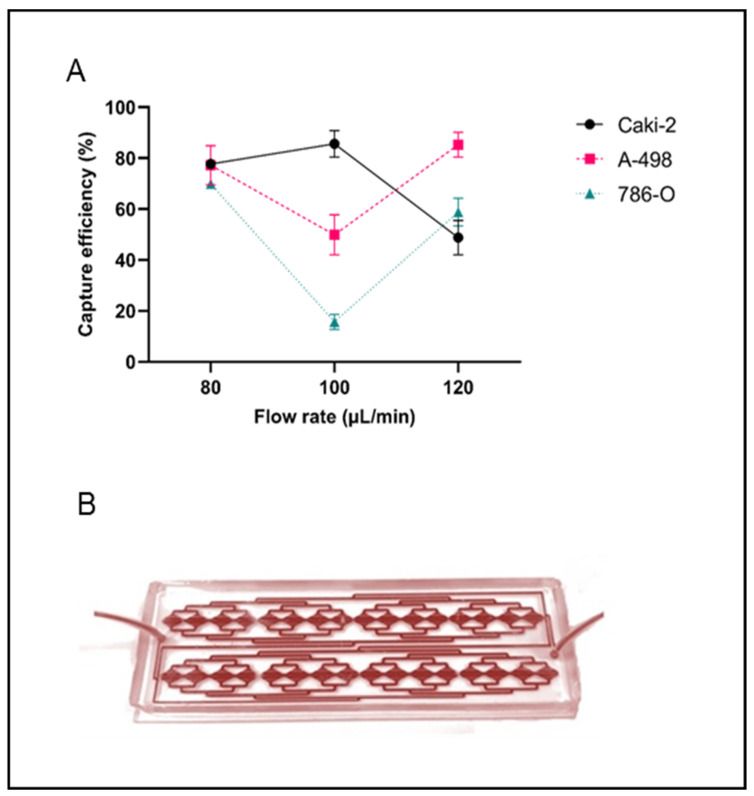
(**A**) Capture efficiency (%) at 80, 100, and 120 μL/min flow rate for Caki-2 (circles with continuous black line), A-498 (squares with dashed red line), and 786-O (triangles with dashed blue line) cells using the RUBYchip^TM^ device. For each flow rate, capture efficiency is represented as the mean and standard deviation (SD) of triplicate experiments. (**B**) RUBYchip™ device running the blood sample of a patient.

**Figure 2 ijms-24-08404-f002:**
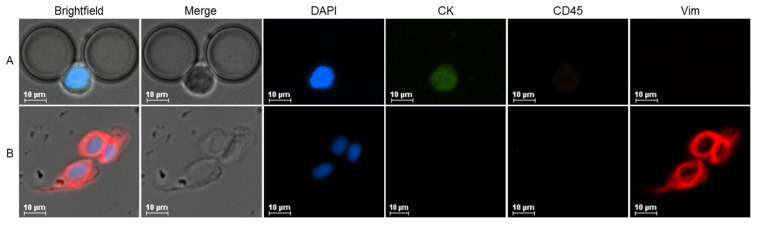
Fluorescence microscopy images at 20X magnification of CTCs from RCC patients captured with RUBYchip^TM^. (**A**) Epithelial CTC (DAPI+/CD45-/CK+/Vim-). (**B**) CTC cluster formed by mesenchymal CTCs (DAPI+/CD45-/CK-/Vim+).

**Figure 3 ijms-24-08404-f003:**
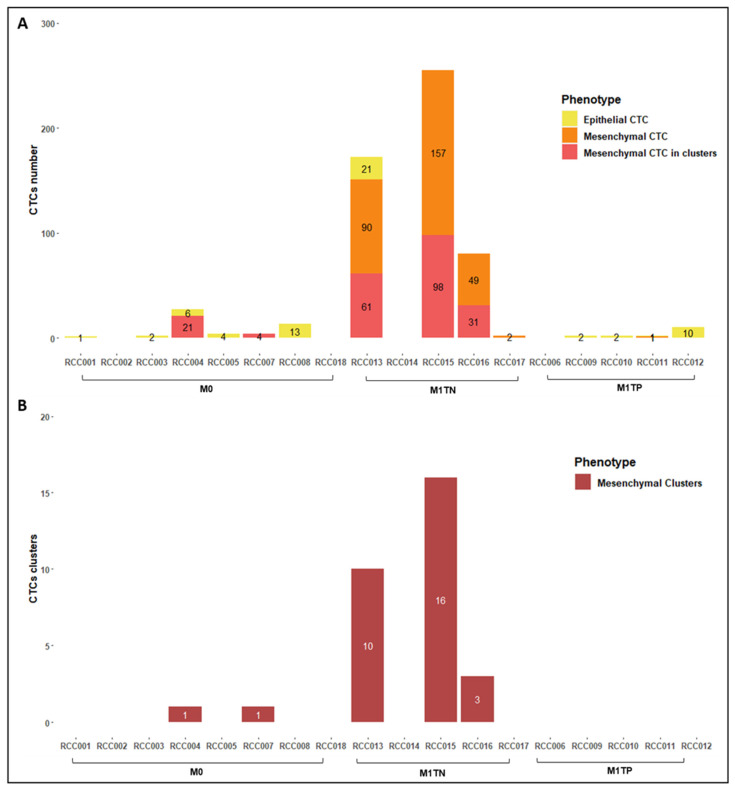
Number of CTCs in M0, M1TN, and M1TP patient groups. (**A**) Number and phenotype of single CTCs and CTCs in clusters. (**B**) Number and phenotype of CTC clusters. Epithelial CTCs are represented in yellow bars, mesenchymal single CTCs in orange bars, mesenchymal CTCs in clusters in light red bars, and mesenchymal clusters in dark red bars.

**Figure 4 ijms-24-08404-f004:**
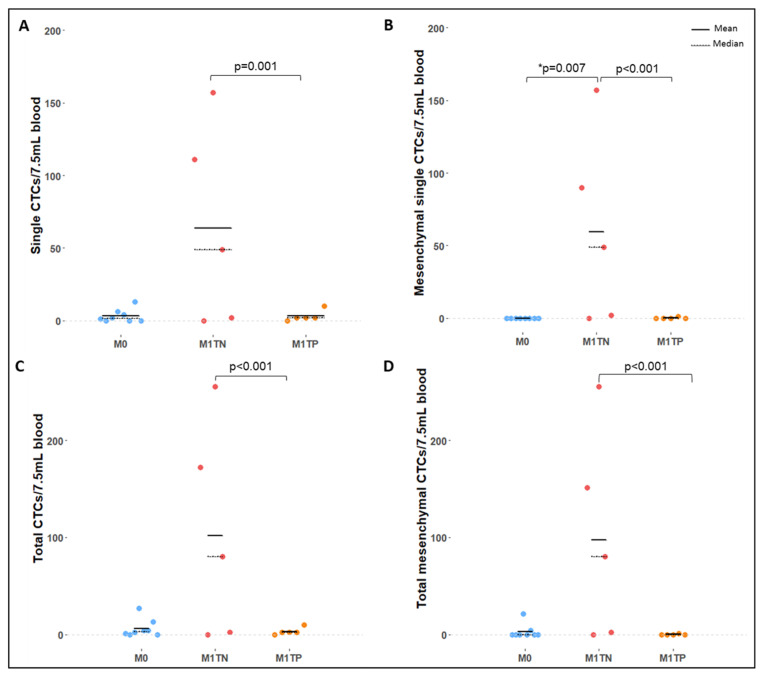
Comparison of CTC counts between M0 (blue dots), M1TN (red dots), and M1TP (orange dots) patient groups. (**A**) Single CTC counts per 7.5 mL whole blood. (**B**) Single mesenchymal CTC counts per 7.5 mL whole blood. (**C**) Total CTC counts per 7.5 mL whole blood. (**D**) Total (single + clustered) mesenchymal CTC counts per 7.5 mL whole blood. Mean represented as a continuous black line and median as a dashed black line. *p*-values obtained via negative binomial regression. * *p*-value obtained through Fisher’s test for presence/absence of CTCs.

**Figure 5 ijms-24-08404-f005:**
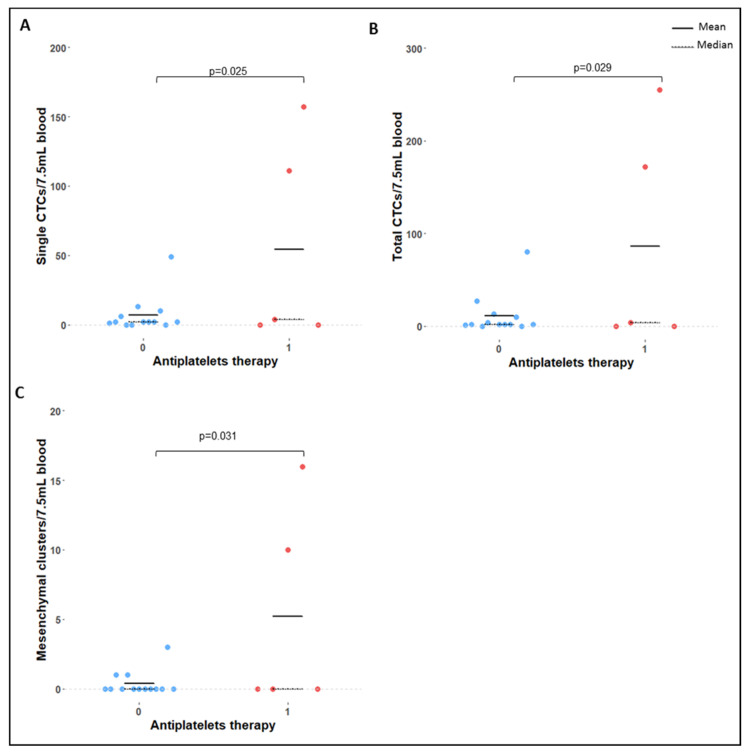
(**A**) Single CTC, (**B**) total CTC, and (**C**) mesenchymal CTC cluster counts in patients. (1) receiving and (0) not receiving antiplatelet therapy. Mean represented as a continuous black line and median as a dashed black line. *p*-values obtained via negative binomial regression.

**Figure 6 ijms-24-08404-f006:**
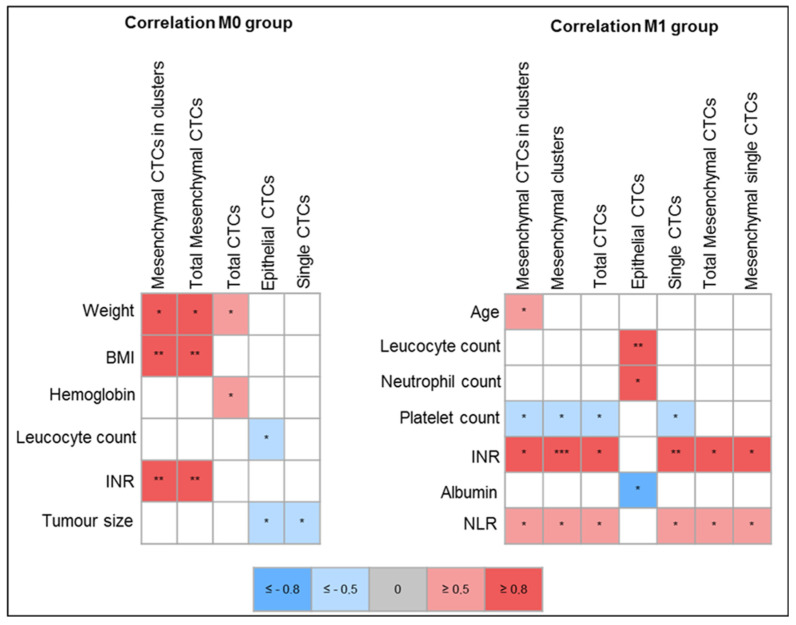
Correlation between CTC counts and clinical variables. BMI—body mass index; CTC—circulating tumor cell; INR—international normalized ratio; M0—localized disease; M1—metastatic disease; NLR—neutrophil-to-lymphocyte ratio. * *p* < 0.05, ** *p* < 0.01, *** *p* < 0.001. *p*-values retrieved from Pearson’s correlation test. Blue depicts negative correlations: light blue—moderate correlation (r ≥ −0.5); strong blue—strong correlation (r ≥ −0.8). Red depicts positive correlations: light red—moderate correlation (r ≥ 0.5); strong red—strong correlation (r ≥ −0.8).

**Figure 7 ijms-24-08404-f007:**
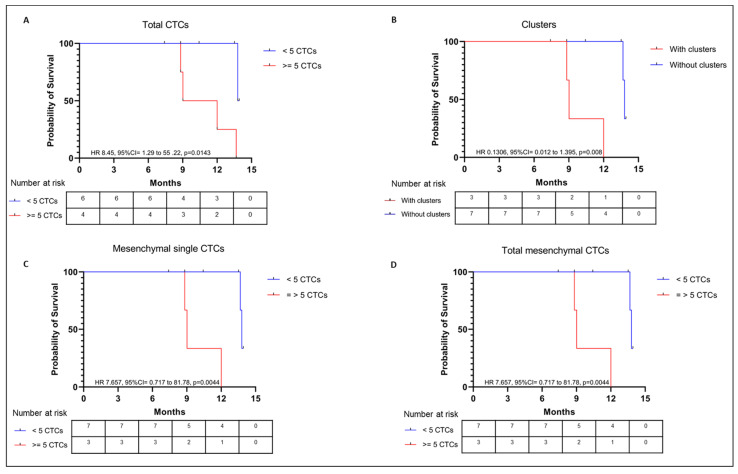
Kaplan-Meyer curves for OS of the metastatic RCC patients’ group regarding (**A**) total CTCs, (**C**) single mesenchymal CTCs, and (**D**) total mesenchymal CTCs <5 (blue line) and ≥5 (red line); and for (**B**) presence (red line) or absence (blue line) of CTC clusters.

**Table 1 ijms-24-08404-t001:** Clinicopathological characteristics of the study patient cohort.

Clinicopathological Characteristics	Overall	M0	M1TN	M1TP	*p*-Value
Number of patients	18	8	5	5	
Gender, n (%)					
Female	5 (28.0)	2 (25.0)	1 (20.0)	2 (40.0)	1
Age, years					
Median (range)	60 (43–78)	60 (52–70)	71 (43–78)	60 (48–69)	0.511
Smoking habits, n (%)	7 (38.9)	5 (62.5)	1 (20.0)	1 (20.0)	0.252
Obesity, n (%)					
Overweight/Obesity	12 (66.7)	7 (87.5)	1 (20.0)	4 (80.0)	0.05
BMI score (%)					
Median (range)	25.5 (18–38.6)	25.5 (23.6–38.6)	21.5 (21.0–25.4)	25.5 (18.0–27.1)	0.049
Hypertension, n (%)	12 (66.7)	6 (75.0)	3 (60.0)	3 (60.0)	1
Diabetes, n (%)	5 (27.8)	3 (37.5)	1 (20.0)	1 (20.0)	1
ECOG score, n (%)					0.384
0	10 (55.6)	4 (50.0)	3 (60.0)	3 (60.0)	
1	4 (22.2)	3 (37.5)	0	1 (20.0)	
2	2 (11.1)	1 (12.5)	0	1 (20.0)	
3	2 (11.1)	0	2 (40.0)	0	
T stage, n (%)					0.003
T1a	6 (33.3)	6 (75.0)	0	0	
T1b	3 (16.7)	2 (25.0)	1 (20.0)	0	
T2a	2 (11.1)	0	1 (20.0)	1 (20.0)	
T2b	2 (11.1)	0	1 (20.0)	1 (20.0)	
T3a	5 (27.8)	0	2 (40.0)	3 (60.0)	
N stage, n (%)					0.045
N0	13 (72.2)	8 (100.0)	3 (60.0)	2 (40.0)	
N1	5 (27.8)	0	2 (40.0)	3 (60.0)	
Histology, n (%)					0.515
Clear cell	8 (72.7)	4 (80.0)	1 (50.0)	3 (75.0)	
Chromophobe	1 (9.1)	0	1 (50.0)	0	
Papillary	2 (18.2)	1 (20.0)	0	1 (25.0)	
No biopsy (patient preference or unfit)	3	3	3	1	
Metastatic site, n (%)					1
Lung	-	-	3	3	
Bone	-	-	1	1	
Distant lymph nodes	-	-	1	1	
Antiplatelet therapy, n (%)	5 (27.8)	3 (37.5)	2 (40.0)	0	0.416
Systemic therapy, n (%)					
First line	-	-	2	2	
Second line	-	-	-	3	
Unfit for treatment	-	-	3	-	
Treatment, n (%)					
TKI	4 (22.2)	-	1 (20.0)	3 (75.0)	
ICI	3 (16.7)	-	1 (20.0)	2 (40.0)	
Radical nephrectomy	8 (72.7)	5 (62.5)	0	-	
Partial nephrectomy	1 (9.1)	1 (20.0)	0	-	
Surveillance	2 (11.1)	2 (25.0)	0	0	
Unfit for treatment	3 (16.7)	0	3 (60.0)	0	

BMI—body mass index; ECOG—Eastern Cooperative Oncology Group; ICI—immune checkpoint inhibitor; M0—localized disease patient group; M1TN—metastatic treatment-naive patient group; M1TP—metastatic progressing-under-treatment patient group; N—node; T—tumor; TKI—tyrosine kinase inhibitor (according to the American Joint Committee on Cancer (AJCC) Cancer Staging Manual Eighth Edition, 2017 [37]). *p*-values concern Fisher’s exact test for categorical variables and Kruskal-Wallis test for the quantitative variables.

**Table 2 ijms-24-08404-t002:** CTC count and phenotype of the study cohort.

	M0			M1TN			M1TP		
	Median	Average	Range	Median	Average	Range	Median	Average	Range
**Single CTCs**	1.5	3.3	0–13	49	63.8	0–157	2	3.2	0–10
Epithelial	3	4.2	0–13	0	5.3	0–21	2	3	0–10
Mesenchymal	0	0.0	—	49	59.6	0–157	0	0.2	0–1
**CTC clusters**	0	0.25	0–1	3	5.8	0–16	0	0.0	—
CTCs in clusters (Mesenchymal)	0	3.1	0–21	31	38.0	0–98	0	0.0	—
**Total CTCs ***	3	6.4	0–27	80	101.8	0–255	2	3.2	0–10
Epithelial	3	4.2	0–13	0	5.3	0–21	2	3	0–10
Mesenchymal	0	3.1	0–21	80	97.6	0–255	0	0.2	0–1

CTC—circulating tumor cell; M0—localized disease patient group; M1TN—metastatic treatment-naive patient group; M1TP—metastatic progressing-under-treatment patient group. * Total CTCs = single CTCs + CTCs in clusters.

## Data Availability

The data presented in this study are available in the Appendix A.

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
