# Peer review of "Clinical Validation of a Size-Based Microfluidic Device for Circulating Tumor Cell Isolation and Analysis in Renal Cell Carcinoma"

_ijms, 2023, doi:10.3390/ijms24098404_

Round 1
Reviewer 1 Report
In their interesting manuscript “Clinical validation of a size-based microfluidic device for circulating tumor cell isolation and analysis in renal cell carcinoma”, authors highlight the clinical importance of biomarkers in renal cancer as they can help in making the clinical decision. They discuss the potential use of CTCs as biomarkers in RCC patients using RUBYchip. The manuscript is well-written and structured and provides promising evidences on the efficiency of RCC-CTC isolation as liquid biopsy. However, there is one issue to be explained. Authors analyze 18 patients, but in the Table 1, the histology is described only for 11 patients. I assume the rest (7 patients) is not known. Nevertheless, the 18, not the 11 patients should be defined as 100%. Please correct it.
Reviewer 2 Report
This study entitled “Clinical validation of a size-based microfluidic device for circulating tumor cell isolation and analysis in renal cell carcinoma “is providing evidence for the isolation of CTCs from RCC patients using a new technology Rubychip. The results are interesting; however, the patient cohort is too small to draw firm conclusions. The authors must also clarify some issues:
· In the abstract, the authors do not clarify how many patients have been analyzed per stage.
· In spiking experiments to evaluate the best flow rate, the authors have only checked one concentration 200 cells/7.5ml which is rather high. Have the authors checked lower concentrations?
· It is strange that in patients in the M1TP stage, the number of CTCs is very low compared to M1TN and the authors must give a better explanation in the Discussion for this fact.
· In Table 2 the authors could also give percentages of CTCs/total CTCs
· The authors must provide a table with all the patients and the absolute number of CTCs per phenotype.
· In Figure 6 there are no correlations in M1PN patients. Are there significant correlations between these patients and other clinical parameters?
· In the survival curves the authors included all the patients despite the stage of the disease. This is a very fragile conclusion, considering the small number of patients. In my opinion, these curves must be in suppl materials.
· The discussion section is quite large. I think it should be smaller to be better for readers.
· I the M&M section the authors explained that they have used two different cytokeratin antibodies. Which is the FITC-conjugated antibody (line 532-533)
Round 2
Reviewer 2 Report
The authors have successfully replay to most of my concerns